# RADPAC-PD: A tool to support healthcare professionals in timely identifying palliative care needs of people with Parkinson's disease

Herma Lennaerts-Kats[1,2]*, Jenny T. van der Steen[3,4], Zefanja Vijftigschild[2], Maxime Steppe[1], Marjan J. Meinders[5], Marten Munneke[1], Bastiaan R. Bloem[1], Kris C. P. Vissers[2], Marieke M. Groot[2]

**1** Department of Neurology, Radboud University Medical Center, Donders Institute for Brain, Cognition and Behavior, Nijmegen, The Netherlands, **2** Department of Anesthesiology, Pain and Palliative Care, Radboud University Medical Center, Nijmegen, The Netherlands, **3** Department of Primary and Community Care, Radboud University Medical Center, Nijmegen, The Netherlands, **4** Department of Public Health and Primary Care, Leiden University Medical Center, Leiden, The Netherlands, **5** Radboud University Medical Center; Radboud Institute for Health Sciences, Scientific Center for Quality of Healthcare, Nijmegen, The Netherlands

* Herma.lennaerts@radboudumc.nl

**Data Availability Statement:** All relevant data are within the paper and its Supporting Information files.

## Abstract

### Background

Parkinson's disease (PD) is a progressive degenerative disease without curative treatment perspectives. Even when palliative care for people with PD seems to be beneficial, the need for palliative care is often not timely recognized.

### Aim

Our aim was to develop a tool that can help healthcare professionals in timely identifying palliative care needs in people with PD.

### Design

We used a mixed-methods design, including individual and focus group interviews and a three-round modified Delphi study with healthcare professionals from a multidisciplinary field.

### Results

Data from the interviews suggested two distinct moments in the progressive PD trajectory: 1) an ultimate moment to initiate Advance Care Planning (ACP); and 2) the actual start of the palliative phase. During the Delphi process, six indicators for ACP were identified, such as presence of frequent falls and first unplanned hospital admission. The start of the palliative phase involved four indicators: 1) personal goals have started to focus on maximization of comfort; 2) care needs have changed; 3) PD drug treatment has become less effective or an increasingly complex regime of drug treatments is needed; and 4) specific PD-symptoms or complications have appeared, such as significant weight loss, recurrent infections, or

**Funding:** This study is funded by The Netherlands Organization for Health Research and Development (ZonMw). (Grant reference number 80-84400-98-086). www.zonmw.nl Prof. Bastiaan R. Bloem was supported by a research grant of the Parkinson's Foundation. The funding party had no role in the design of the study, or in data collection, analysis or in writing the manuscript.

**Competing interests:** The authors have declared that no competing interests exist.

**Abbreviations:** PD, Parkinson's Disease; SPC, Specialist Palliative Care; WHO, World Health Organization; COREQ, Consolidated criteria for reporting qualitative research; GP, General Practitioner.

progressive dysphagia. Indicators for both moments are included in the RADboud indicators for PAlliative Care Needs in Parkinson's Disease (RADPAC-PD) tool.

## Conclusion

The RADPAC-PD may support healthcare professionals in timely initiating palliative care for persons with PD. Identification of one or more indicators can mark the need for ACP or the palliative phase. We expect that applying the RADPAC-PD, for example on an annual basis throughout the PD trajectory, can facilitate identification of the palliative phase in PD patients in daily practice. However, further prospective research is needed on the implementation of the RADPAC-PD.

## Introduction

Parkinson's disease (PD) is a progressive neurological disease that affects physical, psychosocial and spiritual aspects of life [1–3]. Initially, people with PD may experience relatively stable periods, but debilitating symptoms develops over time progressively, many of which do not respond adequately to any medical treatment. PD is viewed as a life-limiting disease, hence a palliative care approach may meet the needs of patients and caregivers [4, 5].

The World Health Organization (WHO) defined palliative care as "an approach that improves the quality-of-life of patients and their families facing problems associated with life-threatening illness, through the prevention and relief of suffering by means of early identification and impeccable assessment and treatment of pain and other problems, physical, psychosocial and spiritual" [6]. Palliative care was initially developed for patients with incurable diseases such as cancer. But palliative care can also be beneficial for patients with a chronic progressive disease trajectory, such as heart failure, dementia and COPD. Meanwhile, palliative care in PD is widely advocated [4, 5, 7, 8]. Early qualitative studies [9–11] have shown that patients with advanced PD have palliative care needs. Late-stage PD is associated with considerable disability that is comparable to that of end-stage cancer [12, 13].

Although the potential need for palliative care is increasingly acknowledged, still little is known about effective and useful components in PD patients [4, 5, 14]. There is no clear conceptualization of palliative care in PD and as a result, many healthcare professionals experience difficulties in providing palliative care [15–18]. A typical general misconception is that palliative care is associated with cancer, terminal phase care or dying. Furthermore, healthcare professionals find it difficult how and when to start with palliative care or when to refer to specialist palliative care services [15–18]. The difficulty of identifying palliative care needs is likely explained by the slowly progressive nature of PD, with often no clear defining moment for starting with palliative care (unlike e.g. cancer where progression is usually much faster and the timing to start palliative care is often clearer) [4, 5]. Earlier studies pointed out that patients with PD mentioned difficulties to access palliative services [4, 5]. Another problem in not recognizing palliatieve care needs is that physicians may hesitate to discuss progression of the disease in early disease stages–as they fear diminishing hope [4]. Also patients have indicated a conflict of needs: wishing to receive more information on disease progression and death, while fearing to receive the information ("wanting but not wanting," "an information tension") [9]. Richfield et al. [4] pointed out an integrated model of care characterized by close collaboration between neurologist and palliative care specialists, from the time of diagnosis onwards. Besides, the US patient and caregivers council and the European Parkinson's Disease

Association (EPDA) recommends palliative care to be available from the time PD is diagnosed [19, 20].

Not recognizing palliative care needs can lead to unnecessary treatments. In studies of non-cancer chronic diseases, patients received fewer drugs for palliation than patients with cancer [21, 22]. Furthermore, as many persons with PD have substantial cognitive impairment, the process of advance care planning (ACP), in which communication about preferred future health and healthcare is pivotal, might start too late. In palliative care, ACP is important for providing information and support tailored to a persons' needs and wishes. In a small study of patients with PD, only a few had had end-of-life care discussions [23]. Furthermore, a majority of people with PD die in a care home or hospital according to a report by Public Health England [24]. However, many patients and their families indicate a preference for dying at home [25].

Generic tools are available to support healthcare professionals in identifying patients who might benefit from palliative care, such as the Supportive & Palliative Care Indicators Tool (SPICT), the Gold Standards Framework (GSF-PIG) and the RADboud indicators for PAlliative Care Needs (RADPAC) [26–28]. Also the use of a single 'surprise question' [29]: 'Would I be surprised if this patient died in the next 12 months?' can help professionals to identify palliative care needs. However, a systematic review on the use of the 'Surprise Question' showed that this tool performs poorly, and even worse in a population of non-cancer illness, as a predictive tool for death [30]. Furthermore, none of these generic tools are been used in patients with PD. Only, a few studies in PD suggest warning signs for unmet palliative care needs that might trigger healthcare professionals to refer a patient with PD to specialized palliative care [4]. These warnings signs include clinical features associated with causes of death, or features such as recurrent aspiration pneumonia, visual hallucinations, regular falls, dementia) [4, 31, 32]. In this study, we aimed to systematically develop a set of indicators using mixed methodology that can help professionals in proactively providing palliative care for persons with PD.

## Methods

We performed a mixed-methods study including ten individual interviews and three focus group interviews with healthcare professionals (step 1), followed by a three-round modified Delphi study to reach consensus on PD-indicators (step 2). Data were collected from September 2016 till May 2018.

### Ethics

Ethical approval was received from the ethics committee Arnhem-Nijmegen [number; 2016–2424]. Healthcare professionals received oral and written information about the study and written and/or oral consent was obtained. The participating healthcare professionals were not compensated for their contributions to this study. All personal data such as names and work location were anonymised. Informed consent was stored securely in a locked cupboard and electronic data (interview and focus group transcripts, quantitative data) was stored in password protected documents on a secure RadboudUMC server.

### Design

**Step 1: Individual and focus group interviews.** Individual and focus group interviews were held to gain insight in palliative care for PD from the perspective of healthcare professionals. An interview guide was developed starting from the research gap and questionnaire results (unpublished data) [33]. The guide was discussed and amended if needed by HL, MS and MG. An expert panel, consisting of three healthcare professionals and two researchers

with experience in either PD, palliative care or both, reviewed this version and adjusted it. The final version consisted of four topics, each comprising multiple open-ended questions (S1 Appendix). We developed one interview guide for both the individual interviews as well as the focus group discussion. We attempted to use the strengths of both methods as we used the individual interviews to solicit individual views and experiences and in contrast, the focus group discussions served to engage the professionals in further in-depth discussions. The discussion format in the focus groups encouraged interaction between group members and elicited discussion about issues that had not emerge in the individual interviews.

Two of the authors (HL or MS) held individual semi-structured interviews with 10 healthcare professionals. Each interview took between 60 and 90 minutes, and was held at the professional's place of work or other preferred location. Thereafter, we conducted three focus group interviews at a central location in the Netherlands, which lasted between 80 and 100 minutes. The sessions were chaired by either MG or MM, assisted by either HL or MS. One served as a moderator who fostered an active and open discussion; the other took notes (S1 Table). There was no prior relationship between the interviewers and the participants; before the start of the interview, only the interviewers' names and occupations were mentioned. The number of healthcare professionals in the focus groups ranged from 8 to 10. Step 1 was performed from September 2016 till March 2017.

**Step 2: A modified three round Delphi study.** A Delphi study was performed consisting of three rounds to achieve consensus on the statements that had emerged from step 1. For all three rounds, online questionnaires were sent using Castor EDC platform to the same panel members. Confidentiality of individual responses was ensured by the processing of coded data. To stimulate discussions about the ACP and palliative care statements, and to avoid a semantic discussion, the panellists had beforehand received a definition of ACP and a model of care goals developed for people with dementia [34, 35]. The ACP definition we used has recently been developed in an international setting and is supported by the European Association for Palliative Care (EAPC) [34].

Round 1 started off with 34 draft statements distributed over four domains: 1) general statements on Advanced Care Planning (ACP); 2) general statements on palliative care in PD; 3) indicators for the ultimate moment to initiatie ACP; 4) indicators for the start of the actual palliative phase. We used a five-point scale to evaluate the statements (1 = strongly disagree— 5 = strongly agree; options 2–4 were not labelled).

In round 2, we included revised statements from round 1 as well as new indicators. In total 52 statements were provided to the panellists. New indicators were based on feedback from the panellists on open-ended items. General statements on ACP and palliative care were also evaluated on the five-point scale. Indicators for timing of ACP and of the palliative phase were all evaluated on a three-point scale: 1 = no signal, 2 = a supportive signal, 3 = an individual (isolated) signal (see Additional files 3 and 4: Statements for Advance Care Planning and Palliative care).

In the third and final round, the respondents rated two sets of indicators and nine additional statements on the same five-point scale as used in round 1. Six statements were newly added to round 3 based on feedback from round 2. Round 1 was open from 9 February to 6 March 2018 (25 days); round 2 from 20 March to 3 April 2018 (16 days); and the final round 3 from 12 April to 15 May 2018 (33 days). In each round, we sent out two reminders to non-responders.

## Selection of participants

**Step 1: Individual and focus group interviews.** We recruited only healthcare professionals from the Dutch ParkinsonNet [36] who met the following inclusion criteria: registered professionals who over the last two years, treated and supported a person with PD who

subsequently died [33]. ParkinsonNet is a Dutch professional healthcare network with nation-wide coverage, consisting of 70 regional networks which encompass healthcare professionals specialized in PD (n = 3,171) [37–39]. The central idea is that persons with PD should be treated preferentially by a small group of selected professionals with a high degree of expertise in PD [40]. All ParkinsonNet healthcare professionals were sent an email in which they were asked if they would be willing to elaborate on the topic of palliative care in PD in a semi-structured interview or focus group interview. Purposive sampling was employed to ensure that the ultimate sample represented a diverse range of healthcare professionals and characteristics such as age, sex, experience and professional background. We invited healthcare professionals who had expressed willingness to participate for either an individual or focus group interview. Standard procedures were employed for obtaining informed consent from the professionals who eventually participated.

**Step 2: A modified three round Delphi study.** For step 2, we recruited healthcare professionals via ParkinsonNet as well as via the Dutch Palliative care networks. Inclusion criteria for participating in this Delphi study were: 1) being a registered healthcare professional; 2) having at least 5 years' of experience in the field of PD or palliative care, and 3) currently active in clinical practice. We sought for three potential types of end-users of the tool: those with specific expertise regarding care for PD patients but not necessarily with palliative care, those with specific expertise on PD and palliative care, and those with expertise on palliative care but not specifically for PD patients. We also ensured that different disciplines relevant to palliative care in PD were represented. A total of 51 professionals agreed to participate, out of the 56 who had been invited.

## Data collection and analysis

**Step 1: Individual and focus group interviews.** All semi-structured interviews and focus group interview were audio-taped and transcribed. We used an inductive analysis, involving the conceptualization of themes from the transcripts. Two researchers (HL & MS) read and reread the transcripts and, by means of using open coding, coded the first four transcripts of the individual interviews. To increase the coding reliability, all transcripts were initially coded by HL and MS separately. Open codes were compared and discussed until an initial codebook was established. Next, all interviews and focus group transcripts were analyzed based on the codebook independently. Codes were discussed, added, modified or merged if necessary. After coding three interviews and two focus group interviews with the codebook, no new codes emerged and data saturation was reached. Afterwards a process of sorting and classifying codes into sub-themes and themes started. When differences in interpretation between researchers remained, a third senior researcher (MG) was consulted. The reliability of the findings was further enhanced by scrutiny from the research team, which included researchers and practicing clinicians. Once consensus was reached, a final set of themes and subthemes was decided upon [41–43]. The software package Atlas Ti-8 supported the qualitative data analysis. We further followed the reporting adhered to the consolidated criteria for reporting qualitative research (S1 File) [44].

**Step 2: A modified three round Delphi study.** The panellists' responses were used to calculate the levels of agreement [45, 46]. Median ratings of each statement were calculated after each round. A percentage of 70 or higher of respondents rating 4 and 5 on the five-point scale was considered as an acceptable level of agreement. If necessary, and based on the feedback from panellists, statements were discussed by the workgroup (HL, JTvdS, MG, ZV) and revised, except for statements with a percentage below 30. In each round, the panellists were invited to give feedback on each statement. The feedback was used to improve readability and phrasing of questions. The research team summarized the feedback and scores from previous

rounds and introduced this information before the second or third round started. The analyses were performed using SPSS (version 25.0).

## Results

A detailed description of characteristics of healthcare professionals in the individual interviews, focus group interviews and Delphi study is presented in S2 Table. Three main themes and seven subthemes were identified based on data analysis of the individual interviews and focus group interviews.

### Unclear concept of palliative care in PD

Although healthcare professionals identified palliative care as an important aspect of their practice, perceptions of palliative care varied. Some healthcare professionals found that palliative care equated to terminal care, while others had a broader perception of palliative care. A misinterpretation of palliative care rendered healthcare professionals reluctant to discuss palliative care with people with PD or other healthcare professionals. Many of them were not familiar with the WHO definition of palliative care [6] Once the WHO definition had been introduced during the course of the interview, it appeared that many healthcare professionals had the general feeling that this was not fitting the care for people with PD from diagnosis onward. Only a few healthcare professionals noted that palliative care starts at the time of diagnosis. However, the majority of healthcare professionals argued that starting palliative care right after a PD diagnosis was not appropriate (Table 1, Q1).

### Discussions about needs and wishes for future care

Many healthcare professionals reported that a realistic conversation about disease stage, prognosis, possible therapeutic options and disadvantages of proposed treatments is important, but often not performed. Furthermore, discussions about future care (and care goals) should be initiated timely, definitely before cognitive impairment or communication problems arise (Table 1, Q2). However, as e.g. cognitive impairment declines gradually, it often seems unpredictable when a person with PD still can or can no more communicate coherently about his or her wishes for future care. Several professionals had experienced that addressing these important conversations with people with PD came too late.

Another important issue raised during individual interviews and focus group discussions was that healthcare professionals often did not talk about palliative care because of the stigma attached to it (Table 1, Q3). Healthcare professionals argued that speaking about future care too early might lead to depressive feelings in persons with PD. Besides, healthcare professionals further mentioned challenges in communicating with people with PD and their families. They felt that there was a need for an open communication style between the person with PD, their families and healthcare professionals. Family dynamics were not always seen in positive terms, while often communication with the person with PD was not possible due to communication problems and cognitive impairment.

### Initiating palliative care

Healthcare professionals mentioned the implementation of palliative care interventions throughout the course of the disease to relieve symptoms. They argued that two specific marking moments in a PD trajectory should be discerned: 1) for initiating ACP discussions; and 2) for starting the actual palliative phase. It was emphasized that the process of ACP should be separated from 'a palliative phase' in order to avoid postponing speaking about ACP to a

**Table 1. Themes and subthemes from the individual interviews and focus group interviews.**

| Main theme | Subthemes | Quote number | Quote |
|---|---|---|---|
| Unclear concept of palliative care in PD | Perceptions of palliative care | Q1 | In principle, I don't think a patient with PD right after diagnosis is per se a "palliative patient". People can still live a very long time with the disease. (individual interview, physiotherapist) |
| | Lack of defining palliative care | | I think it is a difficult question, to talk about palliative care from diagnosis. I don't see it that way. People can live with PD for more than 30 years. On the other hand, PD care and palliative care overlap. The focus in palliative care is on quality of life. But, considering PD, from diagnosis you can only suppress symptoms. PD care is about, how can I live the best possible life with the disease. (individual interview, occupational therapist) |
| Discussions about needs and wishes for future care | Timely speaking about future care | Q2 | Especially in PD, you need to speak timely about what a person's needs are. I often notice that it is too late, that a patient with PD develops severe cognitive problems, so that he cannot speak for himself anymore. In fact, you need to prevent that (focus group interviews) |
| | Stigma | Q3 | It is just the terminology, because I relate palliative care to dying. If a person is just diagnosed with PD, I do not think of the term death. However, I think that the goals of care are much identical (individual interview, nurse practitioner) |
| | Family dynamics | | The question is whether the term palliative is too complicated for people. You should consider the best way to bring the message. Treatment of the disease is not meaningful, comfort and giving meaning to it, that's what is should be about. I don't care about which term you use as a professional. It can be palliative, but I have a different association with this term, more like end-of-life care (focus group interviews, neurologist) |
| Initiating palliative care | Two marking moments<br><br>Need for knowledge on ACP and role and responsibilities | Q4 | This is such a clear and obvious moment: a neurologist says I have not many medical options anymore. Honestly, I think it is fine, but say it. Most patient do know, because they do not see the advantage of going to a neurologist anymore. Patients say "I'll go for a talk but there is little value". In a way, everyone feels that this moment is coming and that there are many PD medical options left. However, doctors don't speak it out loud. I'm convinced that you should speak about this moment and hand over the care of a patient to a GP. (individual interview, elderly care physician) |

moment when the patient him- or herself can no longer be involved. Healthcare professionals defined 'the palliative phase' as a period when they should actively identify palliative care needs or when care goals focuses more on comfort. A number of indicators for both moments were proposed during the individual interviews and the focus group discussions (Table 1, Q4). However, none of them was felt to be a key indicator on its own. Furthermore, health care professionals suggested that not only the knowledge about ACP should be improved, but that it also should be made clearer by whom ACP must be initiated.

## Delphi process

Statements which did reach consensus on ACP and palliative care are presented in S3 Table. Furthermore, results from rounds 1, 2 and 3 are presented in S2 Appendix. Fig 1 shows the final RADPAC-PD tool.

### Statements on palliative care

Our panel suggests that patients with PD are eligible for palliative care and that the support of patients with PD should include palliative care. The statement that PD is an illness you could die from was supported by only half of the panellists (S2 Appendix Table, round 1: statements 15) It was argued that patients do not directly die from PD, but are more likely to die from complications or co-morbidities. Therefore, this statement was rephrased and split into three separate statements (S2 Appendix Table, round 2: statements 13, 14, 15). Furthermore, it was stated that palliative care is multidimensional and pays attention to an individual's wellbeing

Figure 1 RADboud indicators for PAlliative Care Needs in Parkinson's Disease (RADPAC-PD)

With regard to the patient, is there any indication of the following?

**Part 1: Indicators for Advance Care Planning***

1. signals or requests for advance care planning or end-of-life care discussions
2. loses hope or dreads the future
3. frequent falls (resulting in a hip fracture, for example)
4. dysphagia or a first aspiration pneumonia episode
5. cognitive deficits and/or neuropsychiatric problems
6. an (first) unplanned hospital admission

**Part 2: Indicators to identify onset of a patient's palliative phase****

1. preferred goal of care moves towards maximization of comfort
2. a transition in care needs, for example recurrent hospital admissions, nursing home admission and an increase in help of activities of daily living
3. Parkinson's Disease drug treatment less effective or increasingly complex regime of drug treatments
4. several specific Parkinson's Disease symptoms / complications such as significant weight loss, recurrent infections, progressive dysphagia, neuropsychiatric problems and/or multiple falls

\* at least two indicators should be present for initiating ACP
\*\* at least one indicator should be present for the start of the actual palliative phase

**Fig 1. RADPAC-PD.**

in the physical, psychological, social and spiritual domains. It was suggested that professionals should take a patient's need as the leading principle in the decision-making process, even if this is contrary to wishes of the family. With regard to the roles of different professionals, there should be a shared responsibility for identifying the onset of the palliative phase according to our panellists. Timely identification was seen as important, as well as the provision of palliative care in accordance with the patient's needs and wishes for future care.

The participants agreed on indicators for the start of the palliative phase, with two general indicators and two disease-specific indicators. The first referred to the moment when the preferred goal of care moves towards maximization of comfort. Other important indicators related to transitions in care, such as recurrent unplanned hospital admissions, nursing home admission or an increase in ADL-support. Furthermore, an important disease-specific indicator was the moment when PD drug treatment becomes less effective or increasingly complex. Complications such as significant weight loss, recurrent infections, progressive dysphagia, neuropsychiatric problems and multiple falls were identified as PD-specific indicators. There was no agreement on the value of the surprise question as an additional indicator for marking

the onset of the palliative phase, nor on the minimum number of indicators for the RAD-PAC-PD that should be present to identify onset of a patient's palliative phase.

## Statements on ACP

Advance care planning was seen as an important part of optimal palliative care. All respondents agreed with statements that ACP discussions can reduce anxiety and uncertainty. However, only the panellists with expertise in palliative care agreed that information about the prognosis can help patients to prepare better for future decision making. The panellists did not agree about the moment of initiating ACP. Some commented that ACP should be provided shortly after diagnosis. No consensus was reached on the statements about starting ACP at a certain period after diagnosis. Indicators for the initiation of ACP were found to be more helpful. Again, we found general and disease-specific indicators. A person's individual's readiness stage was suggested as an important indicator for panellists to start with ACP. The panellist agreed that all professionals can play a role in facilitating ACP, although a leading role herein was envisaged for the primary treating practitioner. It was mentioned that currently, physicians and nurses lack the required competencies and skills to discuss ACP.

## Discussion

An important new finding from this study is that healthcare professionals suggest two important marking moments in a PD disease trajectory: 1) the ultimate moment to initiate ACP; and 2) the start of the actual palliative phase. Differentiating these two moments might help professionals to discuss needs and wishes of palliative care with a person with PD before he or she looses decisional capacity. Recognizing a palliative phase can contribute to evaluate appropriate care goals that are in line with the actual (medical) situation of a person with PD. Participants emphasized that the process of ACP should start earlier, even before the palliative phase. But this might also be due to the fact that ACP was not seen as part of palliative care by healthcare professionals. Many healthcare professionals emphasized the need for ACP as cognitive problems may already be present even at an earlier stage of PD trajectory [47]. Many of patients with PD will eventually develop dementia [4, 48, 49]. Therefore, professionals emphasized that the process of ACP should be separated from the palliative phase in order to avoid that a person with PD can no longer be involved due to a lack of decision-making capacity and communication problems. The suggested indicators that could contribute to specify both marking moments included dysphagia, dementia, weight loss, and falls, among other things. The indicators were either PD-specific symptoms or general indicators such as recurrent hospital admissions and more support needed for activities of daily living. Some of these indicators have also been described in earlier studies as triggers to initiate palliative care in PD [4, 31, 32].

This study also supports earlier findings that healthcare professionals experience difficulties in the conceptualization of palliative care for people with PD. Only fifty percent of our panellist in the Delphi study agreed on the statement that PD is a disease one can die from. Therefore, we doubt whether healthcare professionals regard PD as a terminal illness. As is known from studies in the field of dementia, healthcare professionals who do recognize a disease as terminal are more likely to initiate palliative care [35, 50]. In this study, many healthcare professionals report different views on palliative care and often equated palliative care with terminal care. Results from the individual and focus group interviews shows that professionals working in PD were not familiar with the WHO definition and more recent insights on palliative care, which was also found in earlier qualitative studies [16–18, 51]. Discussions about what essential elements are in palliative care for people with PD resulted in several specifications, such as 'discussions about future care' and 'care goals focusing on comfort'.

Recognizing a palliative phase can contribute to evaluate appropriate care goals that are in line with the actual (medical) situation of a person with PD. However, different opinions about the time of the palliative phase were identified in the individual and focus group interviews. Participants emphasized that the process of ACP should start earlier, even before the palliative phase. But this might also be due to the fact that ACP was not seen as part of palliative care by healthcare professionals. Many healthcare professionals emphasized the need for ACP as mild cognitive impairment may already be present even at the moment of diagnosis [47]. Averaged across all stages, about a quarter of patients with PD develop dementia [48, 49], but ultimately, most will develop at least some form of dementia [4, 48, 49]. Therefore, professionals emphasized that the process of ACP should be separated from the palliative phase in order to avoid that a person with PD can no longer be involved due to a lack of decision-making capacity and communication problems. The suggested indicators that could contribute to specify both marking moments included dysphagia, dementia, weight loss, and falls, among other things. The indicators were either PD-specific symptoms or general indicators such as recurrent hospital admissions and more support needed for activities of daily living. Some of these indicators have also been described in earlier studies as triggers to initiate palliative care in PD [4, 31, 32].

Our Delphi study aimed to develop a tool that exists of a set of indicators for both marking moments. Finally, we had consensus on six indicators for the ultimate moment to initiate ACP and four indicators that might presume the start of a palliative phase. Furthermore, consensus was reached on eight statements on elements of ACP and seven statements for elements for palliative care.

We found no agreement about timing ACP after diagnosis (in months till years after diagnosis) in our Delphi study. This is in line with findings from a study by Rietjens et al [34] that found that ACP can be initiated at any stage of a person's life, but that it should be more targeted when the health condition worsens. Other studies also show that ACP should be introduced when the individual person is ready [34, 52–54]. Both elements are presented in our set of indicators for initiating ACP. We further developed elements that could be helpful in implementing ACP in practice. As earlier studies [5, 14, 15, 55] describe difficulties in role and responsibility on this topic in PD, we found consensus on statements that give direction on how, what and by whom ACP could be performed. Our findings suggest that healthcare professionals of all disciplines do have a role or are entitled to introduce ACP. But final responsibility on the implementation of ACP rests on the primary treating practitioner. In line with this Rietjens et al [34] described consensus on the fact that a trained non-physician facilitator can support an individual in the ACP process. Van der Steen et al. [35] underlined a multidisciplinary approach to communication and shared decision making in people with dementia. Our results suggest in more detail a responsibility for the primary treating practitioner.

We identified four indicators in the disease trajectory of a person with PD that healthcare professionals considered signals to start the actual palliative phase. We found that all professionals have a role in identifying onset of the palliative phase. We did not found consensus on how many indicators should be present in order to identify the onset of a patients' palliative phase. However, our panellist did agree on the fact that for initiating ACP at least two indicators should be present. We recommend that for the ultimate moment to initiate ACP at least two of six indicators are present. For the start of the actual palliative phase initiating at least one indicator should be present. Applying the RADPAC-PD, for example on an annual basis throughout the PD trajectory, can facilitate identification of PD patients' palliative phase in daily practice. Our study found no consensus on adding the 'surprise question' to the RADPAC-PD. However, earlier research has found that the surprise question combined with an additional screening instrument seems to have a higher validity to identify patients likely in

need of palliative care in specific conditions [26, 28, 56]. Further prospective research is needed on the implementation of the RADPAC-PD.

In this study, healthcare professionals report difficulties in recognizing palliative care needs because of the individual patient's variability in clinical presentations and their rate of disease progression. We therefore suggested to develop a specific PD identification tool which includes some clinical indicators that are equated with the prognostic indicator guide of the Gold Standards Framework (GSF-PIG) and SPICT [26, 28]. Compared to the RADPAC-PD (which includes 5 PD-specific indicators out of 10), the GSF and SPICT include more different indicators relevant to the patient's general health for COPD, heart failure or cancer. An important difference is that these general instruments only provide an indication of the start of the actual palliative phase, while the introduction of the RADPAC-PD suggests two marking moments in a PD disease trajectory of each individual patient.

This study describes the systematic development of the RADPAC-PD based on qualitative research and a Delphi study. We do not yet have evidence about the applicability of this instrument. We acknowledge that further research is needed on the implementation of the RADPAC-PD.

## Strengths & limitations

The high response rate from panellists suggests that palliative care is perceived as relevant to clinical practice. Nevertheless, selection bias cannot be ruled out, with preferential inclusion of professionals who are already interested in providing palliative care. We did not perform member checking with the study participants for the individual and focus group interviews. However, the research team members regularly discussed data, coding and themes throughout analyses. Results from the interviews and focus group discussions were used as input for the Delphi study in which we searched for agreement. We acknowledged bias, maintained neutral, and objectively presented the methods and procedures for enhancing the rigor of research findings in the research process. Furthermore, whether the RADPAC-PD can support professionals in practice in offering palliative care is unknown. In a follow-up study, we will evaluate the usefulness of the RADPAC-PD. Additionally, as more evidence becomes available about its usefulness, we may find new arguments to further adjust the RADPAC-PD. Furthermore, the RADPAC-PD tool was developed in the Dutch medical landscape. International and cultural differences in providing care for patients with PD may affect the usefulness in other countries or other healthcare settings. We do expect that this tool can help healthcare professionals to identify two specific moments in a PD trajectory. The RADPAC-PD tool is a first instrument that separates those two important moments and can be a valuable contribution to the care for people with PD.

## Supporting information

**S1 Table. Researchers' characteristics.**
(DOC)

**S2 Table. Characteristics of participants.**
(DOC)

**S3 Table. Statements and ratings for palliative care and ACP.**
(DOC)

**S1 Appendix. Interview guide for individual interviews and focus group interviews.**
(DOC)

**S2 Appendix. Statements and ratings for palliative care and ACP from round 1, 2, 3.**
(DOCX)

**S1 File. COREQ checklist.**
(PDF)

**S2 File.**
(XLSX)

## Acknowledgments

We thank healthcare professionals who participated in the study for their support of this study.

## Author Contributions

**Conceptualization:** Herma Lennaerts-Kats, Jenny T. van der Steen, Marjan J. Meinders, Marten Munneke, Bastiaan R. Bloem, Kris C. P. Vissers, Marieke M. Groot.

**Data curation:** Herma Lennaerts-Kats, Jenny T. van der Steen, Zefanja Vijftigschild, Kris C. P. Vissers, Marieke M. Groot.

**Formal analysis:** Herma Lennaerts-Kats, Jenny T. van der Steen, Zefanja Vijftigschild, Marjan J. Meinders, Marieke M. Groot.

**Funding acquisition:** Herma Lennaerts-Kats, Marten Munneke, Bastiaan R. Bloem, Kris C. P. Vissers, Marieke M. Groot.

**Investigation:** Herma Lennaerts-Kats, Zefanja Vijftigschild, Maxime Steppe.

**Methodology:** Herma Lennaerts-Kats, Jenny T. van der Steen, Kris C. P. Vissers, Marieke M. Groot.

**Project administration:** Herma Lennaerts-Kats, Marieke M. Groot.

**Software:** Herma Lennaerts-Kats.

**Supervision:** Jenny T. van der Steen, Marjan J. Meinders, Marten Munneke, Bastiaan R. Bloem, Kris C. P. Vissers, Marieke M. Groot.

**Validation:** Herma Lennaerts-Kats, Zefanja Vijftigschild, Marieke M. Groot.

**Visualization:** Herma Lennaerts-Kats, Marieke M. Groot.

**Writing – original draft:** Herma Lennaerts-Kats, Marieke M. Groot.

**Writing – review & editing:** Jenny T. van der Steen, Zefanja Vijftigschild, Maxime Steppe, Marjan J. Meinders, Marten Munneke, Bastiaan R. Bloem, Kris C. P. Vissers.

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
