## [Decision Letter · Decision Letter 0]

6 Dec 2019

PONE-D-19-24988

RADPAC-PD: a tool to support healthcare professionals in timely identifying palliative care needs of people with Parkinson’s disease

PLOS ONE

Dear drs Lennaerts,

Thank you for submitting your manuscript to PLOS ONE. After careful consideration, we feel that it has merit but does not fully meet PLOS ONE’s publication criteria as it currently stands. Therefore, we invite you to submit a revised version of the manuscript that addresses the points raised during the review process.

While both reviewers saw merit in the manuscript, a number of issues were raised. I understand that the authors use a Delphi process, but it does not appear to have been used in practice. I consider whether, in its current form the manuscript will contribute to the base of academic knowledge, which is a requirement for publication. The remaining queries from the reviewers are clearly expressed.

We would appreciate receiving your revised manuscript by Jan 20 2020 11:59PM. To enhance the reproducibility of your results, we recommend that if applicable you deposit your laboratory protocols in protocols.io, where a protocol can be assigned its own identifier (DOI) such that it can be cited independently in the future. For instructions see: http://journals.plos.org/plosone/s/submission-guidelines#loc-laboratory-protocols

We look forward to receiving your revised manuscript.

Kind regards,

César Leal-Costa, Ph. D

Academic Editor

PLOS ONE

Journal Requirements:

Additional Editor Comments (if provided):

I understand that the authors use a Delphi process, but it does not appear to have been used in practice. I consider whether, in its current form the manuscript will contribute to the base of academic knowledge, which is a requirement for publication.

Reviewers' comments:

Reviewer's Responses to Questions

**Comments to the Author**

1. Is the manuscript technically sound, and do the data support the conclusions?

Reviewer #1: Yes

Reviewer #2: Partly

2. Has the statistical analysis been performed appropriately and rigorously? 

Reviewer #1: Yes

Reviewer #2: N/A

3. Have the authors made all data underlying the findings in their manuscript fully available?

Reviewer #1: No

Reviewer #2: Yes

4. Is the manuscript presented in an intelligible fashion and written in standard English?

Reviewer #1: Yes

Reviewer #2: Yes

5. Review Comments to the Author

Reviewer #1: Dear Authors,

I think your paper entitled “RADPAC-PD: a tool to support healthcare professionals in timely identifying palliative care needs of people with Parkinson’s disease” is quite interesting and useful for the whole society. In particular, I think your paper is very useful to improve the activity of health care providers and the welfare of patients and their families.

Parkinson Disease is, like Alzheimer Disease or other neurodegenerative illnesses, one of the critical issues that should be strongly addressed in this century. Both, patients and their families as well as care providers integrate a complex system conditioning the evolution of the disease. Your paper is useful in this process because it can be used to relieve suffering in all the actors involved. Although I think your work deserves to be published, I also think there are some points in the manuscript needing some improvement. As a result, I provide some suggestions here below. I hope my comments and suggestions help you to develop an even better version of your manuscript.

In my view, the mean weakness of the paper is in the theoretical background. I think the introduction is too short. For example, one of the main problems of the introduction is that there is no clear definition of “palliative care”. I think you should provide a theoretical contextualization of the term and related key words (like, for example, “Advanced Care Planning”. Otherwise, misconceptions might arise. Regarding to this, note that in your paper you complain about misconceptions when the transcription from the expert panel is presented (see Table 1, Q1 and Q2). The first sentence in discussion section stresses that idea and, therefore, I think a richer background at introduction will be positive for readers. If you do so, your readers will contextualise better your proposal.

I also find it difficult to assess the usefulness and the impact of the tool you have designed without clearer interpretation rules. For example, the last paragraph in section “Statements of palliative care” addresses that issue but your explanations are not clear for me. When is a practitioner supposed to consider seriously the study of an Advance Care Planning or the beginning of a palliative phase? Is it when one single item has been satisfied? Or, for example, is the number of satisfied items a measure of the urgency for Advance Care Planning or the beginning of a palliative phase? I think these issues deserve a few comments.

I think it would be great if you said whether you compensated the experts for participating in the study (step 1: individual and focus groups interviews). I also think it would be positive if you could provide indices of agreement between experts (for example, percentage of agreement or Cohen’s Kappa coefficients).

There are also some minor issues I would like to give advice about. Firstly, I recommend citing the scientific literature before ending the sentence and not otherwise. For example, in the first sentence the citation is introduced after the dot and I think it should appear before (it appears like: “…. aspects of life.[1-3]”, and I think it is better way: “…. aspects of life [1-3].”).

Secondly, I suggest not using the word “confirmed” in the first sentence of the section “Statements on palliative care”. In my opinion, the sentence with this verb gives the sensation of a strong result or conclusion. In scientific terms things are not so clear sometimes, especially in health and social sciences. Please, consider using another verb like, for example, “agree”.

Finally, I recommend not using abbreviations in Figure 2. I think it will be clearer for readers.

That is all from me, I hope you find my comments and suggestions useful and constructive.

Yours faithfully,

dr46

Reviewer #2: The aim of this study was to develop a tool that can help healthcare professionals in timely identifying palliative care needs in people with PD. Even though the aim can be of interest and it is well written, the manuscript has important problems.

ABSTRACT

- In Methods the authors only present the design. Please include more information about the sample and the analysis methods

INTRODUCTION

It is not clear, in the last paragraph, the need to develop a new specific instrument when instruments already exist, which although generic, include specific sections for the assessment of palliative needs in PD. Which are the problems of these generic tools for their use in PD?.

METHODS

In terms of samples, it is necessary to clarify what the inclusion and exclusion criteria were used for the interviews, focus group and subsequently for the Delphi study.

Related to step 1, it is necessary to clearly separate the methodology by which the interviews were conducted and the methodology by which the focus groups were conducted. For example: the same interview guide was used in the interviews and focus groups, is it not clear?

Related to the analysis of step 1, were transcripts returned to participants? If not, it would be included in limitation section.

There are discrepancies, in the description of step 2 and figure 1, in the number of new indicators which the authors introduce in rounds 2 and 3, they need to be revised. Where do these new indicators come from?, this information is not clear in any section of the article.

Figure 1 is difficult to understand quickly, so it does not do its job well. As far as possible, it should be simplified.

RESULTS

In S4 “Table 2 Characteristics of participants” should include frequency and percentage of participants in each cells.

The analysis in step 1 have important limitations. The authors present the main themes, but they do not describe subthemes neither codes, which are included. As presented, the results do not know which topics were more frequent or were mentioned more in the interviews/focus groups. It would be necessary to include a table with this information. Quotations should be identified by codes, at least if they were identified in interviews or focus groups and placed in quotation marks. In addition, the results do not appear to have been contrasted with participants, which is an important limitation. These questions cast doubt on the credibility of the results.

The presentation of step 2 results appears to be a mere list of statements in no order and confuses the reader. Moreover, as in methods it is not clear how the authors arrive at the set of indicators, the results are very dubious and confusing.

On the other hand, information included S6 Appendix is not mentioned text.

All tables should include the number of participants (n)

DISCUSSION

The authors try to discuss the results, but it is not clear how useful the indicators are, at least those linked to the patient's prognosis, since there are generic instruments for this. However, it seems of greater relevance to have indicators to identify when to start an ACP.

The lack of basic data on applicability and psychometric properties makes it difficult to conclude that these indicators are useful for clinical practice with patients with PD.

REFERENCES

There are errors in the formats of many bibliographic references (e.g., reference 1, 3, 6, 8, etc.). All of them should be reviewed.

6. PLOS authors have the option to publish the peer review history of their article (what does this mean?). If published, this will include your full peer review and any attached files.

Reviewer #1: No

Reviewer #2: No

---

## [Author Response · Author response to Decision Letter 0]

19 Feb 2020

To the Editors of PlosOne

date your letter dated phone fax

February 2020 +3124-3614701 

subject your reference e-mail enclosures 

submission of MS herma.lennaerts@radboudumc.nl

Dear sir Heber,

Thank you for giving us the opportunity to resubmit our manuscript for possible publication in PlosOne. We hereby submit an improved and rewritten version of our manuscript entitled “RADPAC-PD: a tool to support healthcare professionals in timely identifying palliative care needs of people with Parkinson’s disease.” [PONE-D-19-24988]. 

We appreciate the time and effort of the editor and reviewers in a new evaluation of this manuscript. We have revised the manuscript based on the earlier comments. Below, we provide point-by-point responses to these comments and textual proposals for the manuscript. We have uploaded a marked-up copy of the manuscript as well as an unmarked version of the revised paper.

We hope that our revised manuscript can be accepted for publication.

Sincerely,

Also on behalf of the co-authors,

Herma Lennaerts, MSc

Radboud university medical center

Department of Anesthesiology, Pain and Palliative Care

Geert Grooteplein Noord 21 (route 549)

P.O. Box 9101

6500 HB Nijmegen

The Netherlands

Response to the editor 

We are very grateful for the helpful and constructive comments on our manuscript. Below we provide our point-by-point responses.

Editor’s comments

1. While both reviewers saw merit in the manuscript, a number of issues were raised. I understand that the authors use a Delphi process, but it does not appear to have been used in practice. I consider whether, in its current form the manuscript will contribute to the base of academic knowledge, which is a requirement for publication. The remaining queries from the reviewers are clearly expressed.

This study describes how we systematically developed a tool for healthcare professionals which can help to timely provide palliative care. We therefore used a mixed-method design, comprising qualitative research followed by a Delphi study. The qualitative research concerned a thematic analysis to search for patterns and meanings on palliative care for PD. We performed an online Delphi study during the period from February 2018 till May 2018 according to a method described by Linstone et al.[1] We understand that this was insufficiently clear in our original submission, and this is now clarified better in the revision.

Our results contribute new academic knowledge to the field because there is no evidence or tool yet that can help professionals in identifying palliative care needs. Besides results from the Delphi study, further clarify important topics in palliative care (who, when and what) in which no research in PD is available. Furthermore, the tool we developed suggests two specific marking moments in PD rather than a single moment, which is of added value to other generic tools for identifying palliative care needs. The RADPAC-PD in its current form can be used in daily clinical practice, because it is underpinned by both evidence and consensus. Future research can focus on the actual implementation in daily clinical practice. Results of the guidance on ACP and palliative care strategies through the Delphi study can already help in developing palliative care and contribute to academic knowledge. 

Reviewer 1 comments 

1) In my view, the mean weakness of the paper is in the theoretical background. I think the introduction is too short. For example, one of the main problems of the introduction is that there is no clear definition of “palliative care”. I think you should provide a theoretical contextualization of the term and related key words (like, for example, “Advanced Care Planning”. Otherwise, misconceptions might arise. 

We understand your comment, and have now expanded our introduction with a more in-depth theoretical background and definitions of palliative care and ACP. This can be found on page 3, lines 62-113

2) Regarding to this, note that in your paper you complain about misconceptions when the transcription from the expert panel is presented (see Table 1, Q1 and Q2).

The WHO definition does describe palliative care as “an approach that improves the quality-of-life of patients and their families facing problems associated with life-threatening illness, through the prevention and relief of suffering by means of early identification and impeccable assessment and treatment of pain and other problems, physical, psychosocial and spiritual” [2]. Nevertheless, despite the fact that we provided this definition, there is confusion and divergent interpretation of the definition which can induce misconceptions. Our results confirm this notion and are consistent with findings in the field of PD and palliative care [3, 4]. This is now clarified better in the revised version of the paper.

3) The first sentence in discussion section stresses that idea and, therefore, I think a richer background at introduction will be positive for readers. If you do so, your readers will contextualize better your proposal.

We trust that by having added richer background information on palliative care and related terms, better contextualization is possible. We described explicitly which misconceptions we found, as these are the basis of the RADPAC-PD and general statements on palliative care and ACP. We rewrote both the introduction section and the discussion section to better explain the context and background for developing the RADPAC-PD. 

4) I also find it difficult to assess the usefulness and the impact of the tool you have designed without clearer interpretation rules. For example, the last paragraph in section “Statements of palliative care” addresses that issue but your explanations are not clear for me. When is a practitioner supposed to consider seriously the study of an Advance Care Planning or the beginning of a palliative phase? Is it when one single item has been satisfied? Or, for example, is the number of satisfied items a measure of the urgency for Advance Care Planning or the beginning of a palliative phase? I think these issues deserve a few comments.

To provide further information on the usefulness and interpretation rules of the RADPAC -PD, we have now added on page 14-15, lines 378 - 387, the following paragraph: 

We did not find consensus on how many indicators should be present to mark the onset of a patient’s palliative phase. However, the panellists did agree that for initiating ACP, at least two indicators should be present. We recommend to initiate ACP if at least two of six indicators are present. Furthermore, at least one indicator should be present to start the actual palliative phase initiating. Applying the RADPAC-PD, for example on an annual basis throughout the PD trajectory, can facilitate identification of PD patients’ palliative phase in daily practice. Further prospective research is needed on the implementation of the RADPAC-PD. 

5) I think it would be great if you said whether you compensated the experts for participating in the study (step 1: individual and focus groups interviews). 

We added on page 5, line 123, the following sentence: 

The participating healthcare professionals were not compensated for their contributions to this study.

6) I also think it would be positive if you could provide indices of agreement between experts (for example, percentage of agreement or Cohen’s Kappa coefficients).

We now present percentages of agreement for all respondents. The supplement additionally provides agreement percentages for two groups, those with expertise in PD only and those with expertise in palliative care only. We did not calculate Cohens’ Kappa coefficients as our aim was not to calculate inter-rater reliability between single indicators. 

7) There are also some minor issues I would like to give advice about. Firstly, I recommend citing the scientific literature before ending the sentence and not otherwise. For example, in the first sentence the citation is introduced after the dot and I think it should appear before (it appears like: “…. aspects of life.[1-3]”, and I think it is better way: “…. aspects of life [1-3].”).

Thank you; we verified all references and made further adjustments based on criteria of PLOS One. 

8) Secondly, I suggest not using the word “confirmed” in the first sentence of the section “Statements on palliative care”. In my opinion, the sentence with this verb gives the sensation of a strong result or conclusion. In scientific terms things are not so clear sometimes, especially in health and social sciences. Please, consider using another verb like, for example, “agree”.

We agree and have now replaced ‘confirmed’ with the more appropriate scientific term “suggest”. This can be found on page 11, line 283

Our panel suggests that patients with PD are eligible for palliative care and that the management of patients with PD should include palliative care.

9) Finally, I recommend not using abbreviations in Figure 2. I think it will be clearer for readers.

On page 11, Figure 2, we have now described these terms in full. 

Reviewer 2 comments 

10) In Methods the authors only present the design. Please include more information about the sample and the analysis methods

We rewrote our method section and have added more information about the sample and analysis methods. We divided this section into three parts (headed: design, selection of participants, and data analysis) to further enhance comprehensibility. This can be found on page 5 – 8.

11) It is not clear, in the last paragraph, the need to develop a new specific instrument when instruments already exist, which although generic, include specific sections for the assessment of palliative needs in PD. Which are the problems of these generic tools for their use in PD?.

Thank you for raising this topic. On the basis of our qualitative research, we found a need for two specific marking moments, rather than just one. The healthcare professionals emphasized that the process of ACP for people with PD is different from the identification of ‘a palliative phase’. These two moments are important in order to avoid postponing speaking about ACP to a moment when the patient can no longer be involved. The need for two marking moments is specific for PD and is not addressed in the more generic instruments. We developed a new instrument because the participating healthcare professionals mentioned specific PD issues that might be different from, for instance, COPD or heart failure. 

12) In terms of samples, it is necessary to clarify what the inclusion and exclusion criteria were used for the interviews, focus group and subsequently for the Delphi study.

We applied two inclusion criteria for the interviews and focus group discussion: 1) being an active member of ParkinsonNet; 2) being a registered healthcare professional who has treated and supported a person with PD in the past 2 years and who subsequently died. We rephrased the paragraph in which step 1 (page 7, line 178) is described to start with the description of the inclusion criteria. 

We recruited only healthcare professionals from the Dutch ParkinsonNet [5] who met the following inclusion criteria: registered professionals who during the last two years, treated and supported a person with PD who subsequently died [6].

We also clarified the inclusion criteria for the Delphi study on page 7, line 192 by adding the following sentence: 

Inclusion criteria for participating in this Delphi study were: 1) being a registered professional caregiver; 2) having at least 5 years of experience in the field of PD or palliative care and 3) currently active in daily practice.

13) Related to step 1, it is necessary to clearly separate the methodology by which the interviews were conducted and the methodology by which the focus groups were conducted. For example: the same interview guide was used in the interviews and focus groups, is it not clear?

We further clarified individual interviews and focus group interviews at page 5 lines 135 - 140. We added the following text:

We developed a single interview guide for both the individual interviews and the focus group interviews. The individual interviews served to solicit individual views and experiences. In contrast, the focus group discussions served to engage the professionals in further discussions. The discussion format encouraged interaction between group members and elicited discussion about issues that had not emerged from the individual interviews.

14) Related to the analysis of step 1, were transcripts returned to participants? If not, it would be included in limitation section.

No, the transcripts were not returned to individual interviewees and focus group participants. However, the research team members regularly discussed data, coding and themes throughout analyses. We added this limitation on page 15, line 403 – 408:

We did not perform member checking with the study participants for the individual and focus group interviews. However, the research team members regularly discussed data, coding and themes throughout analyses. Results from the interviews and focus group discussions were used as input for the Delphi study in which we searched for agreement. We acknowledged bias, maintained neutral, and objectively presented the methods and procedures for enhancing the rigor of research findings in the research process.

15) There are discrepancies, in the description of step 2 and figure 1, in the number of new indicators which the authors introduce in rounds 2 and 3, they need to be revised. Where do these new indicators come from?, this information is not clear in any section of the article.

We now further describe how new indicators emerged in rounds 2 and 3 in the following section, which is included in the manuscript on page 6, lines 164 – 174:

In round 2, we included revised statements from round 1 as well as new indicators. In total 52 statements were provided to the panellists. New indicators were based on feedback from the panellists on open-ended items. General statements on ACP and palliative care were also evaluated on the five-point scale. Indicators for timing of ACP and of the palliative phase were all evaluated on a three-point scale: 1 = no signal, 2 = a supportive signal, 3 = an individual (isolated) signal (see Additional files 3 and 4: Statements for Advance Care Planning and Palliative care). 

In the third and final round, the respondents rated two sets of indicators and nine additional statements on the same five-point scale as used in round 1. Six statements were newly added to round 3 based on feedback from round 2. Round 1 was open from 9 February to 6 March 2018 (25 days); round 2 from 20 March to 3 April 2018 (16 days); and the final round 3 from 12 April to 15 May 2018 (33 days). In each round, we sent out two reminders to non-responders.

16) Figure 1 is difficult to understand quickly, so it does not do its job well. As far as possible, it should be simplified.

We have now provided the information from the table in the manuscript at page 6, lines 129 – 174, and deleted figure 1.

17) In S4 “Table 2 Characteristics of participants” should include frequency and percentage of participants in each cells.

We adjusted S4 “Table 2 Characteristics of participants” and included the above-mentioned values (marked in yellow).

18) The analysis in step 1 have important limitations. The authors present the main themes, but they do not describe subthemes neither codes, which are included. As presented, the results do not know which topics were more frequent or were mentioned more in the interviews/focus groups. It would be necessary to include a table with this information. 

We have changed Table 1 and have added main themes and subthemes. However, we did not present the frequencies in which themes, subthemes or codes were mentioned. Qualitative data analyzed with a content analysis method can include frequencies at which a topic is mentioned. However, this is not the same for a thematic analysis method in which it is unusual to rate how often a phenomenon is mentioned. A specific item mentioned by only one respondent can be very valuable in understanding the research topic. 

19) Quotations should be identified by codes, at least if they were identified in interviews or focus groups and placed in quotation marks. In addition, the results do not appear to have been contrasted with participants, which is an important limitation. These questions cast doubt on the credibility of the results.

To clarify the origin of the quotes, we included after each quote whether it came from an individual interview or a focus group discussion. Three main themes and seven subthemes emerged after analyzing the qualitative material from part 1. We choose in our first draft not to present all subthemes for reasons of readability. However, we acknowledge that this might confuse readers. We therefore changed table 1. 

The reviewer also mentioned the topic of a member check earlier. We have not discussed the results with participants. However, research team members regularly discussed data, coding and themes throughout analyses. We added this limitation on page 15, line 403 – 408.

20) The presentation of step 2 results appears to be a mere list of statements in no order and confuses the reader. Moreover, as in methods it is not clear how the authors arrive at the set of indicators, the results are very dubious and confusing.

We rewrote our result section in order to improve readability. In the first place, we described at point 15 how new indicators were developed for rounds 2 and 3. Secondly, we present in the manuscript only statements and the final set of indicators that achieve consensus after round 3. Further, in other to clarify how we arrive the set of indicators, we added in the appendix all results from rounds 1, 2 and 3. 

21) On the other hand, information included S6 Appendix is not mentioned text.

Thank you for pointing this out. We added a sentence on page 11, line 279 to introduce S6 Appendix. 

22) All tables should include the number of participants (n)

We added the number of participants to all tables in the manuscript. 

23) The authors try to discuss the results, but it is not clear how useful the indicators are, at least those linked to the patient's prognosis, since there are generic instruments for this. However, it seems of greater relevance to have indicators identifying the moment when to start an ACP.

Based on both reviewers’ comments on the discussion, we rewrote this part and added information about the RADPAC-PD in relation to generic instruments and the need for two marking moments, usefulness of the RADPAC-PD and why it is needed. We rewrote the discussion part at page 15 added following text on the relation of the RADPAC-PD and generic tools (lines 387 – 395):

In this study, healthcare professionals report difficulties in recognizing palliative care needs because of the individual patient`s variability in clinical presentations and their rate of disease progression. We therefore developed a specific PD identification tool to support healthcare professionals. This tool includes clinical indicators that are equated with the prognostic indicator guide of the Gold Standards Framework (GSF-PIG) and SPICT.[7, 8] The GSF and SPICT include more different indicators relevant to the patient’s general health for COPD, heart failure or cancer, than does the RADPAC-PD (which includes 5 PD-disease specific indicators for PD out of 10). Furthermore, while these general instruments only provide an indication of the start of the actual palliative phase, the introduction of the RADPAC-PD suggests two marking moments in a PD disease trajectory of each individual patient.

24) The lack of basic data on applicability and psychometric properties makes it difficult to conclude that these indicators are useful for clinical practice with patients with PD.

Of course, the applicability and psychometric properties of the RADPAC-PD need further prospective investigation. In the current study, we describe the process of development of the RADPAC-PD, an important step in acceptance and use. We might have been too strong in our recommendations about the usefulness for clinical practice without scientific evidence about the applicability. We therefore added the following sentence at the end of the discussion, page 15 lines 396 - 398:

This study describes the systematic development of the RADPAC-PD based on qualitative research and a Delphi study. We do not yet have evidence about the applicability of this instrument. We acknowledge that further research is needed on the implementation of the RADPAC-PD.

25) There are errors in the formats of many bibliographic references (e.g., reference 1, 3, 6, 8, etc.). All of them should be reviewed.

 We reviewed all bibliographic references based on PlosOne criteria policy.

REFERENCES

1. Linstone HA, Turoff M. The Delphi method: techniques and applications. Addison-Wesley, editor1975.

2. WHO definition of palliative care. 2015. http://www.who.int/cancer/palliative/definition/en/ Accessed on 14 May 2015.

3. Waldron M, Kernohan WG, Hasson F, Foster S, Cochrane B. What do social workers think about the palliative care needs of people with Parkinson’s disease? British Journal of Social Work. 2013;43. doi: 10.1093/bjsw/bcr157.

4. Fox S, Cashell A, Kernohan WG, Lynch M, McGlade C, O’Brien T, et al. Interviews with Irish healthcare workers from different disciplines about palliative care for people with Parkinson’s disease: a definite role but uncertainty around terminology and timing. BMC palliative care. 2016;15(1):1-9. doi: 10.1186/s12904-016-0087-6.

5. Bloem BR, Rompen L, Vries NMd, Klink A, Munneke M, Jeurissen P. ParkinsonNet: a low-cost health care innovation with a systems approach from the Netherlands. Health Affairs. 2017;36(11):1987-96.

6. Lennaerts H, Groot M, Steppe M, van der Steen JT, Van den Brand M, van Amelsvoort D, et al. Palliative care for patients with Parkinson's disease: study protocol for a mixed methods study. BMC palliative care. 2017;16(1):61. doi: 10.1186/s12904-017-0248-2. PubMed PMID: 29178865; PubMed Central PMCID: PMCPMC5702094.

7. National Gold Standards Framework Centre England. Gold Standards Framework. http://www.goldstandardsframework.org.uk/evidence (accessed 15 Nov 2018).

8. Highet G, Crawford D, Murray SA, Boyd K. Development and evaluation of the Supportive and Palliative Care Indicators Tool (SPICT): a mixed-methods study. BMJ supportive & palliative care. 2014;4(3):285-90. doi: 10.1136/bmjspcare-2013-000488. PubMed PMID: 24644193.

---

## [Decision Letter · Decision Letter 1]

5 Mar 2020

RADPAC-PD: a tool to support healthcare professionals in timely identifying palliative care needs of people with Parkinson’s Disease

PONE-D-19-24988R1

Dear Dr. Lennaerts,

We are pleased to inform you that your manuscript has been judged scientifically suitable for publication and will be formally accepted for publication once it complies with all outstanding technical requirements.

With kind regards,

César Leal-Costa, Ph. D

Academic Editor

PLOS ONE

Additional Editor Comments (optional):

Reviewers' comments:

Reviewer's Responses to Questions

**Comments to the Author**

1. If the authors have adequately addressed your comments raised in a previous round of review and you feel that this manuscript is now acceptable for publication, you may indicate that here to bypass the “Comments to the Author” section, enter your conflict of interest statement in the “Confidential to Editor” section, and submit your "Accept" recommendation.

Reviewer #1: All comments have been addressed

Reviewer #3: All comments have been addressed

2. Is the manuscript technically sound, and do the data support the conclusions?

Reviewer #1: Yes

Reviewer #3: Yes

3. Has the statistical analysis been performed appropriately and rigorously? 

Reviewer #1: Yes

Reviewer #3: Yes

4. Have the authors made all data underlying the findings in their manuscript fully available?

Reviewer #1: Yes

Reviewer #3: Yes

5. Is the manuscript presented in an intelligible fashion and written in standard English?

Reviewer #1: Yes

Reviewer #3: Yes

6. Review Comments to the Author

Reviewer #1: (No Response)

Reviewer #3: I think you have done an excellent review and a good work.

Finally, I think that you have given a satisfactory answer to the changes previously requested.

7. PLOS authors have the option to publish the peer review history of their article (what does this mean?). If published, this will include your full peer review and any attached files.

Reviewer #1: Yes: Jorge López Puga

Reviewer #3: Yes: José Luis Díaz Agea

---

## [Editor Report · Acceptance letter]

12 Mar 2020

PONE-D-19-24988R1 

RADPAC-PD: a tool to support healthcare professionals in timely identifying palliative care needs of people with Parkinson’s Disease 

Dear Dr. Lennaerts-Kats:

I am pleased to inform you that your manuscript has been deemed suitable for publication in PLOS ONE. Congratulations! Your manuscript is now with our production department. 

With kind regards,

on behalf of

Dr. César Leal-Costa 

Academic Editor

PLOS ONE